# Identifying the Relationship between Leisure Walking and Prevalence of Alzheimer’s Disease and Other Dementias

**DOI:** 10.3390/ijerph19138076

**Published:** 2022-06-30

**Authors:** Junhyoung Kim, Jungjoo Lee, Yu-Sik Kim, Se-Hyuk Park

**Affiliations:** 1Department of Health & Wellness Design, Indiana University, Bloomington, IN 47405, USA; kim9@iu.edu (J.K.); jl174@iu.edu (J.L.); 2Department of Sports Sciences, Seoul National University of Science and Technology, Seoul 01811, Korea; toya00@hanmail.net

**Keywords:** leisure walking, Alzheimer’s disease and dementias, older adults

## Abstract

The literature suggests that leisure walking can play an important role in preventing dementia. The purpose of the present study was to investigate the relationship between leisure walking and the prevalence of Alzheimer’s disease (AD) and other dementias among older adults. Using the 2020 Health and Retirement Study (HRS), 4581 responses constituted the sample for the present study. A hierarchical logit regression analysis was conducted to investigate the relationship between leisure walking and the prevalence of AD and dementia. The results show that leisure walking has been negatively associated with the prevalence of AD and other dementias—that is, they indicate that older adults who frequently engaged in leisure walking were less likely to develop AD and other dementias. This finding suggests the importance of leisure walking as a dementia prevention program for older adults.

## 1. Introduction

Leisure-time physical activity (LTPA) has been advocated as a therapeutic intervention to reduce the risk of developing cognitive decline among older adults [1,2,3]. There is substantial evidence that older adults who are more physically active reported less cognitive decline than those who are less physically active [4,5], suggesting that LTPA plays an important role in preserving and improving the cognitive health of older adults. In addition, multiple longitudinal studies have demonstrated that LTPA is associated with a lower risk of Alzheimer’s disease (AD) and other dementias [6,7,8]. These studies highlight the importance of LTPA engagement as a preventive measure against the development of dementia/AD among older adults.

Among various types of LTPA, a growing body of literature suggests that leisure walking is a highly cost-effective, accessible form of LTPA involving no specific training for preserving the mental and cognitive health of older adults [9,10,11]. Studies focused on the cognitive benefits of walking among older adults have demonstrated that leisure walking is associated with improvements in memory, attention, and functional abilities [12,13].

In a qualitative study, McDuff and Phinney [12] concluded that walking was an effective way for older adults with dementia to be physically active despite their cognitive challenges as well as to prevent the onset or progression of dementia. However, there is a lack of quantitative evidence on the relationship between walking and the prevalence of dementia or AD among older adults. While Abbott et al. [14] found that elderly men who walked frequently were less likely to develop dementia, their study focused only on male participants. There is a need for further investigation of the relationship between walking and dementia including AD among older adults.

While there is no comprehensive conclusion about leisure walking and its effect on specific structural brain changes for older adults, several meta-analyses and clinical studies indicated that physical activity including leisure walking is associated with cognitive functions such as attention, processing speed, and executive function [15,16]. It is assumed that leisure walking can change brain chemistry, making older adults less susceptible to the negative impacts on mental and physical well-being. Thus, the main objective of the present study was to investigate the relationship between leisure walking and the prevalence of AD and other dementias among older adults.

## 2. Method

### 2.1. Data and Study Population

Data comprised a sample drawn from the older adult population (aged 50–90 years) of the 2020 Health and Retirement Study (HRS) COVID-19 Core Early data set. The HRS is a longitudinal study based on a household survey that accumulates information such as demographics, life trajectories (e.g., educational level and income status), health status, and leisure activity participation for older Americans of 50+ years. Data collection for the 2020 HRS was initiated in March 2020 and is still being updated. The 2020 HRS data particularly reflect broad information regarding health status and difficulties related to occupation, leisure behavior, and social support for older adults in the COVID-19 pandemic. This study used secondary data from HRS. Additional approval and consent are not mandatory to work for this study because this study does not fall within the regulatory definition of research involving human subjects.

The latest 2020 HRS includes data from 15,723 households, collected via telephone and email to prevent the COVID-19 infection. From the original data, 4581 responses were extracted and constituted the sample for the present study after correcting missing values on each variable. Table 1 provides an overview of the demographic information of the study sample. The respondents were 39.7% male and 60.3% female and ranged in age from 50 to 98 years old (M = 71.3, SD = 10.4). Over half of the respondents were married (58.5%), 17.8% were divorced, and 15.5% were widowed.

### 2.2. Measurement

Leisure Walking: One item from the 20-item social engagement questionnaire was used to measure leisure walking: “How often do you walk for 20 min or more?” in which respondents answered using a 7-point Likert scale (1 = Daily, 2 = Several times a week, 3 = Once a week, 4 = Several times a month, 5 = At least once a month, 6 = not in the last month, 7 = Never), which was reverse-coded to produce a score that could be compared to the results of Lee et al., (2022). A higher score indicated that respondents engaged in leisure walking more frequently. Table 2 provides an overview of the descriptive statistics for leisure walking (M = 4.52, SD = 2.19). AD was screened by the questionnaire item: “Has a doctor told you that you have Alzheimer’s disease?” to which respondents answered either “Yes” or “No”. Dementia was similarly determined by a questionnaire item: “Has a doctor ever told you that you have dementia, senility, or any other serious memory impairment?” to which respondents answered either “Yes”, or “No”.

### 2.3. Data Analysis

A hierarchical logit regression analysis was conducted to investigate the relationship between leisure walking and the prevalence of AD and dementia. The logistic regression model is used to predict the probability (i.e., odds ratio) of a certain event or status, such as pass/fail, alive/dead, and healthy/sick. The model has been widely used to predict the probability between 0 and 1 of getting a certain type of disease in the future. In this study, the hierarchical logit regression was composed of two blocks. The first block included age and gender as covariates and the second block comprised both block 1 and leisure walking. Before investigating the odds ratio, we conducted an overall model test (Chi-square), the goodness of model fit [17], the model’s explanatory power (Nagelkerke R square), and classification correctness. All statistical analysis procedures were conducted using SPSS 26.0 package. The following is the equation for the hierarchical logit regression model:$$\mathrm{Odds}\;\mathrm{ratio}\;\left(\mathrm{probability}\right)=\frac{\mathrm{onset}\;\mathrm{of}\;\mathrm{disease}}{1-\mathrm{onset}\;\mathrm{of}\;\mathrm{disease}}=\mathrm{Constant}+\mathrm{Age}+\mathrm{Gender}+\mathrm{Leisure}\;\mathrm{Walking}$$
$$\mathrm{Odds}\;\mathrm{ratio}=\beta_{0}+\left\{\left({\mathrm{Age}\;\beta }_{1}+{\mathrm{Gender}\;\beta }_{2}\right)+{\mathrm{Leisure}\;\mathrm{Walking}\;\beta }_{3}\right\}$$

## 3. Results

Table 3 presents the result of the hierarchical logit regression for AD. For an increase in one point of age, the odds ratio of the risk of AD was 1.09 (Wald: 32.91, *p* < 0.05; 95% CI, 1.06–1.12%), and the odds ratio for gender was 0.61 (Wald: 2.83, *p* > 0.05; 95% CI, 0.35–1.08%), meaning that females have a lower risk of AD than males, but the difference was not significant. With the increase in one point for leisure walking, the odds ratio for the risk of AD was decreased to 0.77 (Wald: 13.92, *p* < 0.05; 95% CI, 0.68–0.88%). In other words, the probability of AD decreased by 24% when leisure walking increased by one point. Additionally, the probability of AD increased by 9% when age increased by one point. Following is the equation of the regression model for AD:$$\;\mathrm{The}\;\mathrm{Probability}\;\left(\mathrm{Odds}\;\mathrm{Ratio}\right)\;\mathrm{of}\;\mathrm{Alzheimer}’s\;\mathrm{Disease}=\beta_{0}+1.09{\;\beta }_{1}-0.76{\;\beta }_{3}$$

We conducted an overall model test based on the Chi-square value (64.504, *p* < 0.05). A significant Chi-square presents at least one significant coefficient in the regression model. In other words, there is at least one significant coefficient among age, gender, and leisure walking variables to predict the prevalence of AD. Second, the model explanatory power for the dependent variables (i.e., the probability of AD) was 12.1% based on Nagelkerke R Square (0.121). Smith and McKenna [18] identified that the over 0.4 value of Nagelkerke presents a satisfied model explanatory power. Third, the model for AD has a Hosmer and Lemeshow value of 0.618. A value over 0.05 indicates that a regression model has the goodness of fit (Hu et al., 2006). Finally, the model has a high level of classification correctness (98.9%). In other words, the model predicts the probability of AD with 98.9% correctness. Szumilas [19] and Katz [20] identified that a 70% value of classification correctness is an adequate level to predict the odds ratio.

Table 4 shows the result of the hierarchical logit regression for dementia. With a one-point increase in age, the odds ratio of the risk of dementia was 1.05 (Wald: 23.22, *p* < 0.05; 95% CI, 1.03%–1.07%). The odds ratio of gender was 0.89 (Wald: 0.25, *p* > 0.05; 95% CI, 0.58%–1.38%), which was not significant. In other words, females had a lower risk of dementia than males, but the prediction was not significant in the regression model. With a one-point increase in leisure walking, the odds ratio of the risk of dementia was 0.81 (Wald: 16.40, *p* < 0.05; 95% CI, 0.74–0.90%). In summary, the probability of dementia increased by 5% if age increased by one point. The probability of dementia decreased by 19% if leisure walking increased by one point. The following is the equation for the regression model for dementia:$$\;\mathrm{The}\;\mathrm{Probability}\;\left(\mathrm{Odds}\;\mathrm{Ratio}\right)\;\mathrm{of}\;\mathrm{Dementia}=\beta_{0}+1.05{\;\beta }_{1}-0.81{\;\beta }_{3}$$

This study implemented an overall model test to examine a significant coefficient among independent variables based on a Chi-square value (49.0, *p* < 0.05). The model for dementia has two significant variables (e.g., age and leisure walking). Second, the model’s explanatory power for variances of dependent variables was identified as 6.2% (Nagelkerke R Square). Third, the regression model for dementia was assessed and found to have goodness of model fit (Hosmer and Lemeshow: 0.618). Lastly, the model has a higher-level classification of correctness (98.1%) than the average value (70%). The model predicts the probability of dementia with 98.1% correctness.

## 4. Discussion

In the present study, the relationship between leisure walking and the prevalence of AD and other dementias among older adults was investigated. The results show that leisure walking has been negatively associated with the prevalence of AD and other dementias. They indicate that older adults who frequently engaged in leisure walking were less likely to develop AD and other dementias. This finding suggests the importance of leisure walking as a dementia prevention program for older adults.

In previous studies leisure walking has been recommended as the most cost-effective, accessible form of LTPA as it does not require special training for participation [9,10]. They suggest that a leisure walking group can encourage older adults to gain the mental and cognitive health benefits provided by this form of LTPA [21,22]. The result of the present study is aligned with these studies by showing that leisure walking predicted low prevalence of AD and other dementias in a large sample of older adults.

Similar to the findings of Abbott et al. [14], older adults who frequently walk were less likely to develop dementia and strongly recommended leisure walking for older adults who are at high risk for dementia. The present study expands the idea that leisure walking is instrumental in reducing dementia including AD among older adults. While Abbott’s study focused only on elderly men, our study included both genders and provides evidence that gender did not predict AD and other dementias; however, our study provides evidence that leisure walking has a negative association with the prevalence of AD and other dementias.

The present study expands the body of knowledge on the benefits of leisure walking for older adults’ mental health by suggesting that the amount of leisure walking can contribute to the cognitive health of older adults [23]. In a recent study, Han et al. [9] found that different intensity levels of leisure walking are positively associated with the mental health and health perceptions of older adults. They suggested that moderate and/or vigorous leisure walking can be more beneficial than light leisure walking for older adults’ mental health and health perceptions. In addition, most studies have investigated the benefits of leisure walking for mental health such as reducing depressive symptoms and increasing positive emotions. Our study demonstrates that leisure walking can not only increase mental health and positive health perceptions, but also can reduce the risk of the onset or progression of AD and other dementias. Additionally, this study suggests that the more leisure walking in which older adults participate, the more effective it may be as an intervention for promoting the cognitive health of older adults.

There are some limitations in our study. First, this research is based on a cross-sectional design and has limited power to validate a cause-and-effect relationship between leisure walking and prevalence of AD and other dementias. Longitudinal studies of this relationship are needed. Second, individuals’ functional limitations (e.g., physical capabilities and quality of gait) associated with the aging process can be additional variables that affect leisure walking. Due to the nature of secondary data analysis, our study did not consider individuals’ functional abilities, and future studies are needed to investigate how functional abilities are associated with leisure walking and mental health. Last, the potentially important other variables may include the walking environment and different types of leisure walking which might be examined in the future studies. For example, the effects of solitary walking may be different from those of group walking on the prevalence of Alzheimer’s disease and other dementias.

## 5. Conclusions

Despite some limitations, the present study is an initial investigation of how leisure walking is associated with the prevalence of AD and other dementias among older adults. Overall, leisure walking can be an important therapeutic program for older adults to reduce the risk of the onset or progression of AD and other dementias, suggesting that healthcare providers and researchers need to design and implement effective community-based walking programs for older adults so that they can reduce the risk of developing AD and other dementias. Additionally, the findings of this study expand the body of relevant literature by demonstrating the importance of leisure walking for reducing the prevalence of AD and dementia. Thus, this study indicates that leisure walking can serve as a dementia prevention program for older adults.

## Figures and Tables

**Table 1 ijerph-19-08076-t001:** Characteristics of the study population.

Characteristics	Mean	SD	*n*	%
Age	71.3	10.4		
50 to 98 years old			4581	100
Gender				
Male			1818	39.7
Female			2763	60.3
Marital status				
Married			2679	58.5
Living with a partner			26	1.8
Separated			11	0.2
Divorced			815	17.8
Widowed			711	15.5
Never married			286	6.2

Total *n* = 4581.

**Table 2 ijerph-19-08076-t002:** Descriptive statistics of leisure walking, Alzheimer’s disease, and dementia.

Variables		
Independent Variable	Mean	SD
Leisure Walking	4.52	2.19
Dependent Variables	Onset	None
Alzheimer’s Disease	88	4440
Dementia	51	4529

Total *n* = 4581.

**Table 3 ijerph-19-08076-t003:** Hierarchical Logit Regression for Alzheimer’s Disease.

Variables	B	Wald	Exp(B*)*	CI for Exp(B)
Age	0.85	32.91	1.09 *	1.06–1.12
Gender	−0.48	2.83	0.61	0.35–1.08
Leisure Walking	−0.25	13.92	0.76 *	0.68–0.88
Constant	−9.31	49.12	0.00	
Model test (i.e., Chi-square)	64.504 *			
Model explanatory power (i.e., Nagelkerke R Square)	0.121			
Goodness of model fit (i.e., Hosmer and Lemeshow)	0.618			
Classification correctness	98.9%			

* *p* < 0.05.

**Table 4 ijerph-19-08076-t004:** Hierarchical logit regression for dementia.

Variables	B	Wald	Exp(B*)*	CI for Exp(B)
Age	0.05	23.22	1.05 *	1.03–1.07
Gender	−0.11	0.25	0.89	0.58–1.38
Leisure Walking	−0.20	16.40	0.81 *	0.74–0.90
Constant	−6.71	51.50	0.01	
Model test (i.e., Chi-square)	49.0 *			
Model explanatory power (i.e., Nagelkerke R Square)	0.062			
Goodness of model fit (i.e., Hosmer and Lemeshow)	0.68			
Classification correctness	98.1%			

* *p* < 0.05.

## Data Availability

Health and Retirement Study (HRS) data at https://hrs.isr.umich.edu/data-products.

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
