# Peer review of "Identifying the Relationship between Leisure Walking and Prevalence of Alzheimer’s Disease and Other Dementias"

_ijerph, 2022, doi:10.3390/ijerph19138076_

Round 1
Reviewer 1 Report
I confirm that the points I previously pointed out to you have been thoroughly reviewed and accurately corrected.
The author has fully explained "walking in leisure time." I have also checked the definition of the term.
I have confirmed the author's perspective on the research regarding this study. I also confirmed that the discussion on the quality of walking had been fully considered.
Author Response
Thank you very much of your helpful comments and corrections on the earlier manuscript. We revised the issues raised by the reviewers. Again, thank you so much! We greatly appreciate of your invaluable comments!

Reviewer 2 Report
Thanks for this second version. My main concern in the first review round was about the richness of selection of variables. Now that you have added an explanation in the Methods that you used secondary HRS data, you can now also provide a better explanation about the variable selection, take that change! And pleas expand the Discussion how future research may or should include richer variables.
The titles of the tables are sometimes not very clear and are being duplicatesd in the first line of the table. Example: table 1 is named Demographics, followed by Charecteristics in the first line. Skip that and formulate a more meaningfull title. Suggestion: Table 1. Characteristics of the study population (N=4581). The same for other tables.
The References are numbered twice, please correct that.
Author Response
Thank you very much of your helpful comments and corrections on the earlier manuscript. We revised the issues raised by the reviewers. Again, thank you so much!
Response to Reviewer 2
- Your Comments: “Thanks for this second version. My main concern in the first review round was about the richness of selection of variables. Now that you have added an explanation in the Methods that you used secondary HRS data, you can now also provide a better explanation about the variable selection, take that change! And pleas expand the Discussion how future research may or should include richer variables.”
Answer 1: That is a good point. We explained the variable selection in relation to study limitations in the discussion section as follows: Third, the potentially important other variables may include the walking environment and different types of leisure walking which might be examined in the future studies. For example, the effects of solitary walking may be different from those of group walking on the prevalence of Alzheimer’s disease and other dementias.
- Your Comments: “The titles of the tables are sometimes not very clear and are being duplicated in the first line of the table. Example: table 1 is named Demographics, followed by Characteristics in the first line. Skip that and formulate a more meaningful title. Suggestion: Table 1. Characteristics of the study population (N=4581). The same for other tables.”
Answer 2: Great! We revised as you suggested as follows:
Table 1. Characteristics of the study population
Table 2. Descriptive Statistics of Leisure Walking, Alzheimer’s Disease, and Dementia
- Your Comments: “The References are numbered twice, please correct that.”
Answer 2: Yes! They are corrected!!
We greatly appreciate of your invaluable comments!

This manuscript is a resubmission of an earlier submission. The following is a list of the peer review reports and author responses from that submission.
Round 1
Reviewer 1 Report
Reviewer Report
This cross-sectional study investigates the relationship between leisure walking and the prevalence of Alzheimer's disease and other dementias in the elderly. The research methodology utilizes and analyzes a subset of surveys from a large-scale questionnaire survey. Although this is a cross-sectional study design, the results suggest that older adults who engage in frequent leisure walking are less likely to develop Alzheimer's disease and other dementias.
This study is a critical study that suggests the potential of leisure walking as a program for dementia prevention or treatment.
Major Comments
The author should provide a clear definition of "leisure walking," as suggested in this study. They are linking the question "[How often do you] walk for 20 minutes or more?" included in the survey utilized in this study, the 2020 Health and Retirement Study (HRS) COVID-19 Core Survey, to "leisure time walking" should also be clearly explained. It should also be clearly defined.
An essential aspect of this study is "leisure time," which is also included in the paper's title. The importance of this study will be increased by clarifying the meaning of the theme of "walking in leisure time" rather than "walking."
"Leisure walking" in this study was examined using a 7-point Likert scale. The survey results were related to the frequency of walking for 20 minutes or more. In other words, the study used quantitative findings, i.e., frequency of implementation. In the limitations section of the study, the author describes the method of walking (solo or in a group). Although the quality of gait was not investigated in this study, the quality of pace (speed of gait, the environment in which rate is performed, etc.) can be an essential perspective.
In light of the above, the author should also provide some perspective on the quality of gait in this study.
Reviewer 2 Report
First of all, I would like to thank the editors for the opportunity to review this manuscript, which is eligible for publication in the IJERPH journal.
- "Investigating" is not a proper word for a scientific manuscript.
- I recommend not using abbreviations in the abstract, especially if they have not been explained before. For example AD.
- In the justification, there is a paragraph that makes no sense in this part of the manuscript (from lines 48 to 53). I think it does not correspond to the justification of the study.
- In the tables the abbreviations should come as a footer.
In general, the article meets the standards of being published, but it is very linear, we respond to the relationship between leisure walking and dementia. I have failed to relate it to other variables, and thus reach important conclusions, since there is scientific evidence that relates this chronic disease to many variables.
Reviewer 3 Report
I'm vary sorry to be quite blunt: this is an oversimplified design with a rather poor selection of factors with inappropriate cross sectional data that not add much value to the extensive literature (like Livingston e.a. 2018, 2020).
Reviewer 4 Report
Abstract
State the purpose of the study clearly
Describe the data collection method, age, walking time and how the diagnosis of dementia was made
Results point out the values ​​of the analyzes
Introduction
Point out what the study brings that is different from the others, what is the relevance and the gap in the literature that the study seeks to remedy.
Bring neuroscience explanation to support the study, this is fundamental to justify its relevance
Methods
The tables must be presented in the results
Explain the selection criteria for the elderly
There is a great evaluation bias in the diagnosis of Alzheimer's and dementias, seeking to improve these, because the way the bias is invalidating the findings
Indicate the analysis program
Indicate ethics committee approval
Results
Look for associations with relevant data for dementia, such as: genetics, occupation, depression, among others.
discussion
Address the findings suggested in the new analysis.
Conclusion
Rewrite after reparsing.